DATA RELEASE

# Data from Entomological Collections of *Aedes* (Diptera: Culicidae) in a post-epidemic area of Chikungunya, City of Kinshasa, Democratic Republic of Congo

Victoire Nsabatien[1,2,*], Josue Zanga[2], Fiacre Agossa[3,4], Nono Mvuama[2], Maxwell Bamba[2], Osée Mansiangi[2], Leon Mbashi[2], Vanessa Mvudi[2], Glodie Diza[2], Dorcas Kantin[2], Narcisse Basosila[5,6], Hyacinthe Lukoki[3,7], Arsene Bokulu[3], Christelle Bosulu[3], Erick Bukaka[8], Jonas Nagahuedi[8], Jean Claude Palata[1] and Emery Metelo[3,8]

1 Laboratory of Applied Animal Ecology, Department of Life Sciences, Faculty of Science and Technology, University of Kinshasa, DR Congo
2 Laboratory of Bioecology and Vector Control, Department of Environmental Health, Kinshasa School of Public Health, DR Congo
3 Unit of Entomology, Department of Parasitology, National Institute of Biomedical Research, Kinshasa, DR Congo
4 PMI VectorLink Project, Abt Associates, 6130 Executive Blvd, Rockville, MD 20852, USA
5 Laboratory of Ethnology and Medical Photochemistry, Department of Life Sciences, Faculty of Science and Technology, University of Kinshasa, DR Congo
6 National Malaria Control Program, DR Congo
7 Laboratory of Botany, Systematics and Plant Ecology, Department of Life Sciences, Faculty of Science and Technology, University of Kinshasa, DR Congo
8 Unit of Entomology, Department of Life Sciences, Faculty of Science and Technology, University of Kinshasa, DR Congo

**Submitted:** 18 September 2023

* Corresponding author. E-mail: vnsabatien@gmail.com

Preprint submitted at https://doi.org/10.1101/2023.09.20.558445

Included in the series: ***Vectors of human disease*** (https://doi.org/10.46471/GIGABYTE_SERIES_0002)

## ABSTRACT

Arbovirus epidemics (chikungunya, dengue, West Nile fever, yellow fever and zika) are a growing threat in African areas where *Aedes* (Stegomyia) *aegypti* (Linnaeus, 1762) and *Aedes albopictus* (Skuse, 1895) are present. The lack of comprehensive sampling of these two vectors limits our understanding of their propagation dynamics in areas at risk of arboviruses. Here, we collected 6,943 observations (both larval and human capture) of *Ae. aegypti* and *Ae. albopictus* between 2020 and 2022. The study was carried out in the Vallee de la Funa, a post-epidemic zone in the city of Kinshasa, Democratic Republic of Congo. Our results provide important information for future basic and advanced studies on the ecology and phenology of these vectors, as well as on vector dynamics after a post-epidemic period. The data from this study are published in the public domain as the Darwin Core Archive in the Global Biodiversity Information Facility.

**Subjects** Ecology, Biodiversity, Taxonomy

## DATA DESCRIPTION
### Background and context

Arbovirus epidemics, particularly chikungunya, are a growing threat in African areas where *Aedes aegypti* and *Aedes albopictus* are present [1–4]. The incomplete sampling of

**Figure 1.** Map of the Funa Valley, showing entomological collection points (North, South, East and West).

these two vectors limits our understanding of their propagation dynamics in areas at risk from arboviruses. Thus, studies such as ours have contributed to understanding the dynamics of these vectors.

Here we describe, for the first time, the geographical distribution of two arbovirus vectors (*Ae. aegypti* and *Ae. albopictus*) in a chikungunya post-epidemic zone in the provincial city of Kinshasa, Democratic Republic of Congo, between 2020 and 2022.

## METHODS

### General spatial coverage

This study was carried out in the Vallée de la Funa (Figure 1), a forest gallery in the hilly area of the commune of Mont Ngafula. It is located within a secondary forest island dominated by *Millettia laurentii* and *Pentaclethra eetveldeana* trees.

The forest gallery is located between latitude 4°25′06.17″S and longitude 15°18′05.77″E, covering around 45 hectares, see Figure 2 for a map of the georeferenced occurrences in GBIF. The average annual rainfall in the area is around 1,095 mm. However, rainfall in Kinshasa is abundant and unevenly distributed throughout the year. The highest volume of precipitation occurs in November, with an average of 268.1 mm, and the lowest volume occurs in July, with an average of 0.7 mm. The average annual relative humidity is 79%.

The mapping of the sampling points for this study was generated using ArcGIS version 10.8 software (RRID:SCR_011081) [5], based on geographical coordinates (east longitude and south latitude) recorded at each entomological collection point using a Garmin Etrex GPmap 64s GPS. The vector layers used on the map were derived from the Référentiel Géographique Commun database of the Democratic Republic of Congo.

### Mosquito collection

The general taxonomic coverage description for this work is the Culicidae Family, specifically *Ae. aegypti* (commonly known as yellow fever mosquito; NCBI:txid7159) and *Ae. albopictus* (commonly known as the Asian tiger mosquito or moustique tigre in French; NCBI:txid7160).

Two sampling techniques were used to collect immature and adult stages of *Aedes* spp. between January 5, 2020–December 30, 2022. Whatever the collection method, morphological identification of adult mosquitoes was carried out using the taxonomic keys [6].

### Larval collection

During the 2020–2022 study period, larvae were collected once a year in October, which heralds the return of heavy rains after the dry season stretching from mid-May to mid-August. Immature stages of *Aedes* spp. were collected from various breeding sites (abandoned pots, tires, cans and other containers) using the dipping technique. In this technique, water is taken from the breeding sites using a ladle or a small container, and it is used to search for Culicidae larvae, specifically those of *Aedes* spp. These larvae were collected and stored in jars containing water from their respective breeding sites, then transported to the insectarium of the Laboratory of Bioecology and Vector Control (or BIOLAV), where they were reared until adulthood under insectarium conditions (temperature: $28 \pm 1\,°C$; relative humidity: 70–80%; photoperiod 14 h:10 h light:dark). In the insectarium, larvae were placed in rearing tanks according to the type of larval habitat they were found in the field; pupae pipetting and adult emergence followed the same classification procedures. Five days after emergence, adult mosquitoes were sucked into cages, placed in jars and cooled to $-23\,°C$ to prevent their escape during identification. The identification data was encoded into an Excel sheet, and the mosquito samples were stored at $-23\,°C$ for potential future analysis.

### Human landing catches

Following ethical approval and consent, the host-seeking *Aedes* collections were carried out over four samplings each year, for three years, and distributed as follows: January "short rainy season accompanied by a short dry season"; June "dry season"; August "early rainy season"; October "rainy season". During each sampling campaign, collectors were supervised inside and outside each house to ensure the smooth running of capture activities. Supervisors responded to collectors' concerns and made unannounced visits to each capture station to ensure the work's quality was in accordance with the instructions.

Human landing catches (HLC) were used to collect *Aedes* spp. adults during the day [7]. At the study site, mosquito collections were carried out both indoors and outdoors, with a first group of collectors working from 7:00 a.m. to 12:00 p.m. and replaced by a second group from 12:00 p.m. to 7:00 p.m. At each collection point, a bare-legged, barefoot volunteer served as bait, collecting mosquitoes using hemolysis tubes. Mosquito samples were then transported to the morphological identification room of the Bioecology and Vector Control Laboratory for morphological identification.

### Quality control description

Mosquitoes were identified using keys made available in the literature [6] by an entomologist experienced in the identification of Central African *Aedes*. Once digitized,

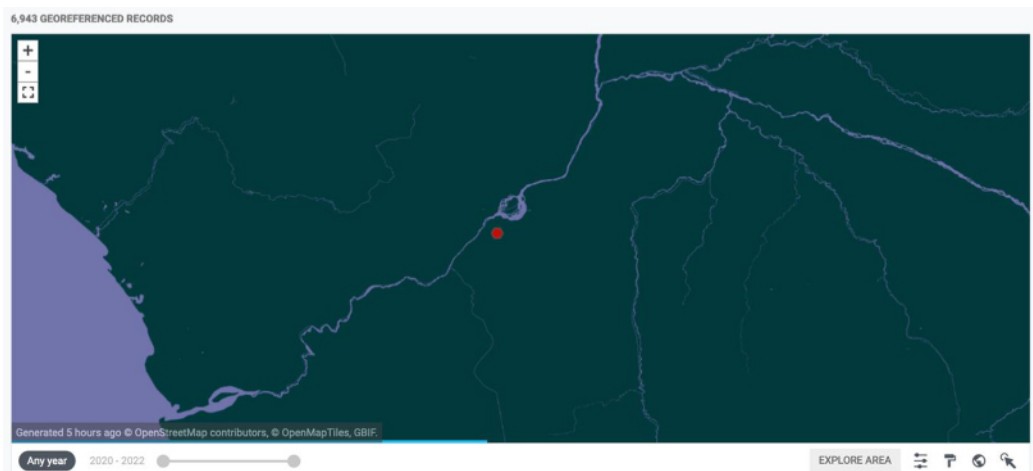

**Figure 2.** Interactive map of the georeferenced occurrences hosted by GBIF [8]. https://www.gbif.org/dataset/438cb683-7d8e-4677-bc30-013b675973fc

**Table 1.** Summary of the results obtained using two entomological sampling techniques after three years of study in a Chikungunya post-epidemic area.

| Year Collected | Entomological sampling used | Season | *Ae. aegypti* | | *Ae. albopictus* | | *Aedes spp.* | | Total | |
|---|---|---|---|---|---|---|---|---|---|---|
| | | | (n) | (%) | (n) | (%) | (n) | (%) | (n) | (%) |
| 2020 | HLC | Rain | 255 | 39.7 | 269 | 41.8 | 0 | 0 | 643 | 100.0 |
| | | Dry | 2 | 0.3 | 117 | 18.2 | 0 | 0 | | |
| | Larva collect | Rain | 156 | 28.8 | 385 | 71.2 | 0 | 0 | 541 | 100.0 |
| 2021 | HLC | Rain | 66 | 3 | 2,008 | 92.4 | 0 | 0 | 2,172 | 100.0 |
| | | Dry | 9 | 0.4 | 82 | 3.8 | 7 | 0.3 | | |
| | Larva collect | Rain | 0 | 0 | 665 | 100 | 0 | 0 | 665 | 100.0 |
| 2022 | HLC | Rain | 49 | 2.7 | 1,608 | 88.2 | 0 | 0 | 1,823 | 100.0 |
| | | Dry | 8 | 0.4 | 158 | 8.7 | 0 | 0 | | |
| | Larva collect | Rain | 33 | 3 | 1066 | 97 | 0 | 0 | 1,099 | 100.0 |
| 3-yearly | HLC | **Rain** | **370** | **7.9** | **3,885** | **83.9** | **0** | **0** | **4,638** | **100.0** |
| | | **Dry** | **19** | **0.4** | **357** | **7.7** | **7** | **0.2** | | |
| | **Larva collect** | **Rain** | **189** | **8.2** | **2,116** | **91.8** | **0** | **0** | **2,305** | **100.0** |

the data was validated using the Integrated Publishing Toolkit (IPT) validator tool available from the Global Biodiversity Information Facility (GBIF) [8].

## RESULTS

A total of 6,943 observations (summarized in Table 1) were reported using larval capture and HLC methods between 2020 and 2022 (Figure 2). The HLC collecting method was responsible for the collection of 4,638 adult mosquitoes, while the larval sampling yielded 2,305 individuals. *Ae. albopictus* was the most abundant species in the studied region, with 6,358 individuals, whereas we recorded 578 *Ae. aegypti* individuals. Only seven individuals could not be identified to the species level.

## RE-USE POTENTIAL

The results of this study will provide important information for future basic and advanced studies on the ecology and phenology of these vectors, as well as on vector dynamics after an epidemic period.



Given that these vectors have been responsible for a considerable disease burden during the various Chikungunya epidemic periods in the Democratic Republic of Congo, the results of these entomological surveys can help national policy-makers target appropriate interventions and allocate resources to national programs for vector control. Our results may also help explain why existing interventions are not working as intended.

## DATA AVAILABILITY
The data supporting this article are published through the IPT of the University of Kinshasa and are available via GBIF under a CC0 waiver [8].

## DATASET DESCRIPTION
**Object name:** Darwin Core Archive Data from Entomological Collections of *Aedes* (Diptera: Culicidae) in a post-epidemic area of Chikungunya, City of Kinshasa, Democratic Republic of Congo
**Character encoding:** UTF-8
**Format name:** Darwin Core Archive format
**Format version:** 1.0
**Distribution:** https://cloud.gbif.org/africa/archive.do?r=aedes_drc
**Publication date of data:** 2023-06-30
**Language:** English
**Licences of use:** Public Domain (CC0 1.0)

**Metadata language:** English
**Date of metadata creation:** 2023-06-30
**Hierarchy level:** Dataset

## EDITOR'S NOTE
This paper is part of a series of Data Release articles working with GBIF and supported by TDR, the Special Programme for Research and Training in Tropical Diseases hosted at the World Health Organization [9].

## ABBREVIATIONS
BIOLAV, Laboratory of Bioecology and Vector Control; GBIF, Global Biodiversity Information Facility; HLC, human landing catches; IPT, Integrated Publishing Toolkit.

## DECLARATIONS
### Ethical approval and consent to participate
Ethical approval for this study was obtained from the Ethics Committee of the School of Public Health, Kinshasa (University of Kinshasa), Ref: ESP/CE/91/2020. Meetings were held with heads of households and catchers, where the purpose and procedures of the study were explained. In particular, permission was sought from heads of households to inspect mosquito larval habitats to sample larvae and adult mosquitoes in their respective households.

### Competing Interests
The authors declare that they have no competing interests.

## Authors' contributions

EM and JZ contributed to the design and coordination of the project, as well as the quality control of the mosquito samples. FA contributed to the coordination and provision of resources for the operational execution of this study. VN contributed to data curation, visualization, formal analysis on excel, and drafting of the manuscript. VN, NM, VM, OM, MB, LM, GD, DK, HL, AB and CB supervised the field study, conducted laboratory experiments, including mosquito sorting and morphological identifications. EB, NB, JP and JN critically revised the manuscript. All authors read and approved the final manuscript.

## Authors' information

JZ, NM, VN, VM, OM, MB, LM, GD and DK are involved in vector surveillance and control, and ITNs durability at national level, and are all researchers in the Bioecology and Vector Control Laboratory at the Kinshasa School of Public Health (University of Kinshasa). EM, FA, HL, AB and CB, carry out vector mapping and resistance monitoring activities at national level, and are all researchers in the Entomology Unit of the National Institute of Biomedical Research. JN and EB are conducting studies on larval control and mapping of Culicidae breeding sites in the city of Kinshasa, and are researchers in the Entomology Unit at the University of Kinshasa. VN and JP are conducting studies on the population ecology of arthropods of medical interest, and are researchers in the Applied Animal Ecology Laboratory at the University of Kinshasa. NB is conducting studies on the mapping of Anopheles vectors of malaria at national level and monitoring invasive species, and is the focal point for the National Malaria Control Programme. FA is also chef of party for the PMI Evolve project in the DRC.

## Funding

The study was self-financed by the Bioecology and Vector Control Laboratory based at the Kinshasa School of Public Health.

## Acknowledgements

The authors would like to thank the Bioecology and Vector Control Laboratory at Kinshasa School of Public Health for providing the facilities to colonize the larval stages of *Aedes* until emergence, and the instruments required for mosquito identification. In addition, many thanks to Dear Willy LUSASI for his willingness to design the map of the Funa valley, showing the entomological collection points for this study. We would also like to thank the communities of the Funa Valley for allowing us into their homes to collect samples. We would also like to thank Paloma Helena Fernandes Shimabukuro and Tsiky Rabetrano for their help with data submission and facilitation in using the GBIF platform.

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
