## [Editor Report]

Editor’s AssessmentAedes mosquito spread Arbovirus epidemics (e.g. Chikungunya, dengue, West Nile, Yellow Fever, and Zika), are a growing threat in Africa but a lack of vector data limits our ability to understand their propagation dynamics. This work describes the geographical distribution of Ae. aegypti and Ae. albopictus in Kinshasa, Democratic Republic of Congo between 2020 and 2022. Sharing 6,943 observations under a CC0 waiver as a Darwin Core archive in the University of Kinshasa GBIF database. Review improved the metadata by adding more accurate date information, and this data can provide important information for further basic and advanced studies on the ecology and phenology of these vectors in West Africa.

---

## [Reviewer Report]

Upload additional filesDRR-202309-01/form/DRR-202309-01_Data-Review-MAT.pdfReviewer name and names of any other individual's who aided in reviewer Mary Ann TuliDo you understand and agree to our policy of having open and named reviews, and having your review included with the published papers. (If no, please inform the editor that you cannot review this manuscript.)YesIs the language of sufficient quality?YesPlease add additional comments on language quality to clarify if needed
n/aAre all data available and do they match the descriptions in the paper? YesAdditional CommentsnoneAre the data and metadata consistent with relevant minimum information or reporting standards? See GigaDB checklists for examples <a href="http://gigadb.org/site/guide" target="_blank">http://gigadb.org/site/guide</a>YesAdditional CommentsIs the data acquisition clear, complete and methodologically sound?YesAdditional CommentsIs there sufficient detail in the methods and data-processing steps to allow reproduction?YesAdditional CommentsIs there sufficient data validation and statistical analyses of data quality? YesAdditional CommentsIs the validation suitable for this type of data?YesAdditional CommentsIs there sufficient information for others to reuse this dataset or integrate it with other data?YesAdditional CommentsAny Additional Overall Comments to the AuthorRecommendationAccept

---

## [Reviewer Report]

Reviewer name and names of any other individual's who aided in reviewer Paul TaconetDo you understand and agree to our policy of having open and named reviews, and having your review included with the published papers. (If no, please inform the editor that you cannot review this manuscript.)YesIs the language of sufficient quality?YesPlease add additional comments on language quality to clarify if needed
Some minor changes that I recommend : "And the relative annual average humidity is 79%." may be changed to "The relative annual average humidity is 79%.". "Aedes albopictus is the most abundant species in the studied region" may be changed to "Aedes albopictus was the most abundant species in the studied region"Are all data available and do they match the descriptions in the paper? NoAdditional Comments1/The data available are of type 'occurrence' ( only in 1 file - the "occurrence" file). For a better presentation of the data, I would suggest to transform them into "sampling event" data, which is more suited to this kind of data acquired from sampling events (see https://ipt.gbif.org/manual/en/ipt/latest/sampling-event-data), while keeping the occurrence dataset. This would enable the user to quickly understand the dates and locations of the sampling events.  2/ In the data, the only available date (column eventDate) is the first of January (eg. 2021-01-01T00:00:00). This does not enable to separte the data into seasons (Rainy et Dry) as presented in table 1 of the manuscript. I strongly suggest the authors to provide the specific date for each collected mosquito in the data.
Are the data and metadata consistent with relevant minimum information or reporting standards? See GigaDB checklists for examples <a href="http://gigadb.org/site/guide" target="_blank">http://gigadb.org/site/guide</a>YesAdditional CommentsIs the data acquisition clear, complete and methodologically sound?NoAdditional Comments1/Larval collections : sampling strategy used ?  2/How many collection rounds in total ? please provide the dates of collection.
Is there sufficient detail in the methods and data-processing steps to allow reproduction?YesAdditional CommentsIs there sufficient data validation and statistical analyses of data quality? NoAdditional Comments1/Human landing catch : was any quality control done during the collection of data (i.e. check that the collectors were at their place, etc.) ? Is the validation suitable for this type of data?YesAdditional CommentsIs there sufficient information for others to reuse this dataset or integrate it with other data?YesAdditional Comments1/comments for figure 1 (map) :  - "legend" should be written in english (and not in french) - "harvesting sites" -> entomological collection points - the background layer is not very appropriate. Maybe better to put an Open Street Map background layer  2/What about ethical approval for the Human Landing Catches ? please provide the name of the institution who has approved the HLC and the approval number, if relevant  3/ in the dataset, for the species scientific name, I suggest to use the names as presented in : Harbach, R.E. 2013. Mosquito Taxonomic Inventory, https://mosquito-taxonomic-inventory.myspecies.info/ . Or at least, to provide the "nameAccordingTo" column.  4/ In the dataset, many columns seem totally empty. Please remove them if so.  
Any Additional Overall Comments to the AuthorThanks for this nice work and the effort put to publish your entomological data. I strongly suggest you to add the real dates of collection of the data in the GBIF dataset (see comments above).RecommendationMajor Revision

---

## [Reviewer Report]

Reviewer name and names of any other individual's who aided in reviewer Angeliki F MartinouDo you understand and agree to our policy of having open and named reviews, and having your review included with the published papers. (If no, please inform the editor that you cannot review this manuscript.)YesIs the language of sufficient quality?YesPlease add additional comments on language quality to clarify if needed
Are all data available and do they match the descriptions in the paper? YesAdditional CommentsIt will be good for the authors the first time that they cite the two species to use the full names Aedes (Stegomyia) albopictus (Skuse) Aedes (Stegomyia) aegypti (Linnaeus, 1762)  In the methods section the title should be Human Landing Catches and not Human capture on landingAre the data and metadata consistent with relevant minimum information or reporting standards? See GigaDB checklists for examples <a href="http://gigadb.org/site/guide" target="_blank">http://gigadb.org/site/guide</a>YesAdditional CommentsIs the data acquisition clear, complete and methodologically sound?YesAdditional CommentsIs there sufficient detail in the methods and data-processing steps to allow reproduction?YesAdditional CommentsIs there sufficient data validation and statistical analyses of data quality? YesAdditional CommentsIs the validation suitable for this type of data?YesAdditional CommentsIs there sufficient information for others to reuse this dataset or integrate it with other data?YesAdditional CommentsAny Additional Overall Comments to the AuthorRecommendationAccept

---

## [Reviewer Report]

Reviewer name and names of any other individual's who aided in reviewer Luis Acuna-CantilloDo you understand and agree to our policy of having open and named reviews, and having your review included with the published papers. (If no, please inform the editor that you cannot review this manuscript.)YesIs the language of sufficient quality?YesPlease add additional comments on language quality to clarify if needed
No applied Are all data available and do they match the descriptions in the paper? YesAdditional CommentsNo appliedAre the data and metadata consistent with relevant minimum information or reporting standards? See GigaDB checklists for examples <a href="http://gigadb.org/site/guide" target="_blank">http://gigadb.org/site/guide</a>YesAdditional CommentsThey must be review the standard Darwin core format for sampling events. https://www.gbif.org/darwin-core.Is the data acquisition clear, complete and methodologically sound?YesAdditional CommentsNo appliedIs there sufficient detail in the methods and data-processing steps to allow reproduction?NoAdditional Comments They don't describe how the map of the study area was created, whether they used a GIS or not. Sampling points must be included on the map.  They don't mention how the identification of the larval stages was carried out and how they were differentiated from other genera of species of the Culicinae subfamily, such as Culex, Haemagogus, Mansonia, Sabethes or other species of the genus Aedes, since the two main species of this genus, were its objective.  In 5 reference, they mention is only for adult identification. They should include or cite the collection protocols and describe them as much as possible so that the study can be replicated in other African countries.Is there sufficient data validation and statistical analyses of data quality? Not my area of expertiseAdditional CommentsThe data could be validated with biological collection of specimensIs the validation suitable for this type of data?NoAdditional CommentsI don't know, but i believed that data could be validated with biological collection of specimensIs there sufficient information for others to reuse this dataset or integrate it with other data?YesAdditional CommentsThe scientific names must follow the same nomenclature, the first time the full name Aedes aegypti is mentioned and the second time Ae.aegypti, if there are two species within the same genus only one is mentioned the first time and the second time both abbreviated Ae.aegypti and Ae.albopictus.  Bibliographic references should be cited accordingly, for example: (1-4).  The names of the diseases must follow the same writing with a capital letter at the beginning or all in lower case Chikungunya or chikungunya.  From the description of the study and the collection times, I would believe that it fits more with Sampling Events, the data is well organized, however, it is suggested to review the Darwin Core template for this type of data and adjust to the corresponding model. , event_core review: https://www.gbif.org/darwin-core.Any Additional Overall Comments to the AuthorThe data paper can be published with suggestions for improvement. Congratulations, very good job!RecommendationMajor Revision